# Radioprotective Activity and Preliminary Mechanisms of *N*-oxalyl-d-phenylalanine (NOFD) In Vitro

**DOI:** 10.3390/ijms20010037

**Published:** 2018-12-21

**Authors:** Yuanyuan Meng, Fujun Yang, Wei Long, Wenqing Xu

**Affiliations:** Tianjin Key Laboratory of Radiation Medicine and Molecular Nuclear Medicine, Institute of Radiation Medicine, Peking Union Medical College and Chinese Academy of Medical Science, Tianjin 300192, China; xiaofanmm@outlook.com (Y.M.); yangfj32@126.com (F.Y.); longwei@irm-cams.ac.cn (W.L.)

**Keywords:** *N*-oxalyl-d-phenylalanine (NOFD), hypoxia inducible factor (HIF), factor inhibiting HIF (FIH), reactive oxygen species (ROS), DNA double-strand breads (DSBs)

## Abstract

The radiation-induced damage to the human body is primarily caused by excessive reactive oxygen species (ROS) production after irradiation. Therefore, the removal of the increase of ROS caused by ionizing radiation (IR) has been the focus of research on radiation damage protective agents. Hypoxia inducible factor (HIF) is a transcription factor in human and plays an important role in regulating the body metabolism. Factor inhibiting HIF (FIH) is an endogenous inhibitor factor of HIF protein under normoxia conditions. It has been shown that the high expression of HIF protein has a certain repair effect on radiation-induced intestinal injury and hematopoietic system damage in mice; however, it is not clear about the effect of HIF on the level of ROS after radiation. In this study, the role of *N*-oxalyl-d-phenylalanine (NOFD), an FIH inhibitor, for its effect on alleviating ROS level is investigated in the cells. Our results indicate that pretreatment with NOFD can mitigate ROS level and alleviate IR-induced DNA damage and apoptosis in vitro. Therefore, HIF can be used as a target on scavengers. Furthermore, in order to explore the relevant mechanism, we also test the expression of relevant HIF downstream genes in the cells, finding that *Notch-2* gene is more sensitive to NOFD treatment. This experiment result is used to support the subsequent mechanism experiments.

## 1. Introduction

Recently, radiation therapy is increasingly used in physical and medical fields, which is also the important treatment of malignant tumors and severe diseases [1]. However, side effects induced by radiation, such as hematopoietic system damage and gastrointestinal dysfunction [2], can have a tremendous potential threat to the human body. Existing research has proven that ionizing radiation (IR) produces a large amount of reactive oxygen species (ROS) directly in the body, mainly through the radiolysis of water [3,4]. Excessive ROS in the body can cause DNA double-strand breaks (DSBs) and increase intracellular oxidative stress and apoptosis [5,6,7,8]. Although most of the research targeted on the scavenging of ROS have been carried out, and so many scavengers were founded and synthesized by researchers, there were few effective drugs available for clinical use limited by their undesirable side effects [9,10,11,12,13,14]. At this point, researchers need to discover new field and compounds exerting protective effect against IR-induced damage.

Hypoxia inducible factor (HIF) protein is a transcription factor in the body, and participates in the regulation of many important metabolic pathways, which plays an important role in maintaining the body’s homostates [15]. Recently, research on HIF protein as the target of radioprotection agents on the ionizing irradiation-induced gastrointestinal dysfunction and hematopoietic system injury had been carried out by two groups [16,17]. HIF protein includes basic helix–loop–helix PAS domain and is composed of heterogeneous dimer comprised with α subunit (HIFα) and β subunit (HIFβ) [15]. Within the body, there are two endogenous inhibitors against HIF protein, prolyl hydroxylase domain (PHD) enzymes and factor inhibiting HIF (FIH) [15]. Under normoxia conditions, HIF-α subunit is hydroxyled on proline and aspartic acid by two endogenous inhibitors separately, and then HIF-α subunit is rapidly targeted for ubiquitin dependent proteasomal degradation by the E3 ubiquitin ligase, termed von Hipple–Lindau disease tumor suppressor (pVHL), or lost the ability to bind to CBP/p300 in the nucleus [15,18,19]. Two enzyme activities are oxygen dependent, so that the transcriptional activity of HIF protein is regulated by oxygen concentration. In normal conditions, both different hydroxylations will cause the HIF protein to lose its transcriptional activity. Based on existing research, a lot of findings supported the role of HIF-1α in reducing the release of ROS in mitochondrion by inhibiting metabolic regulation under mitochondrial respiration, according to increasing the expression of glycolytic enzymes to facilitate anaerobic respiration [20,21,22,23,24]. Another experiment also suggested that the increased mitochondrial ROS induced by hypoxia could be significantly inhibited in FIH-deficient cells [25]. Therefore, we can make an inference that HIF protein can be used as a new target for ROS scavenger research.

In this study, we first synthesized *N*-oxalyl-d-phenylalanine (NOFD), an FIH inhibitor which could block the FIH enzyme ability and increase the transcriptional activity of HIF protein. Next, we evaluated the radiation protection effect of NOFD in vitro. In this experiment, we used cells exposed to 6Gy IR to evaluate the related indicators. According to the results, we found that NOFD exerted protective effect by reducing ROS levels and alleviating the DNA damage and apoptosis in the cells after irradiation. Furthermore, in order to evaluate the potent mechanisms of NOFD removal of ROS in cells, we examined the expression of relevant HIF downstream genes. The results showed that NOFD significantly increased the expression of downstream gene *Notch2*. This provides the basis for our next mechanism research.

## 2. Result

### 2.1. NOFD Reduces Intracellular ROS Levels after Irradiation

ROS are mostly created in the mitochondria during the expression of oxidative phosphorylation by the respiratory chain [26]. In order to detect the effect of NOFD on ROS levels after irradiation in cells, the dichloro-dihydro-fluorescein diacetate (DCFH-DA) assay was used in this experiment. After irradiation, the cells will quickly produce excessive ROS; in order to accurately evaluate the effect of NOFD, the cells after irradiation were immediately collected for treatment and subjected to DCFH-DA staining according to the instructions, and the results were presented with a fluorescence microscope. As shown in Figure 1, compared with the control cells, HCT116 and CHO-K1 cells which only received 6Gy irradiation had significant fluorescent absorption after staining with DCFH-DA. However, the cells administrated NOFD before irradiation showed a weaker fluorescence signal than that in the irradiated cells, suggesting that NOFD could reduce the level of elevated ROS caused by IR in the cells.

### 2.2. NOFD Reduces DSBs after Irradiation in Cells

As reported previously, IR can induce persistent oxidative stress and DNA double-strand breaks (DSBs) damage [27]. Histone H2AX (γ-H2AX) is an important marker of DSBs. In our study, we use the γ-H2AX fluorescent signal method to investigate the damage of IR on DSBs [28,29]. To evaluate whether NOFD could alleviate DSBs induced by IR, we examined the fluorescence signal of γ-H2AX in both CHO-K1 and HCT116 cells. The cells have a self-repairing function, and DNA damage caused by irradiation is also repaired to some extent by the cells, so we chose to examine the damage test 1 h after 6Gy irradiation [30]. Images in Figure 2a,b showed DAPI signal (blue spots), γ-H2AX signals (green spots), and merge signal (γ-H2AX signals on the DAPI background). Compared with the control cells, the irradiated cells exhibited an increase in γ-H2AX immunofluorescence 1 h after IR. However, the cells treated with NOFD before irradiation significantly revealed the decrease of γ-H2AX foci fluorescence compared with that in the irradiated cells. These results suggested that NOFD could reduce DSBs damage induced by IR in the cells.

### 2.3. Comet Assay for DNA Damage in the Cells after Irradiation

In the field of radiation protection research, comet assay is an effective indicator and method for assessing DNA damage after irradiation [30]. In this experiment, we used this marker to assess the effect of NOFD on DSBs. During the experiment, we evaluated the percentage of DNA in tail (Tail DNA%) and tail moment (TM) to further assess IR-induced DNA damage [30]. As described above, HCT116 cells and CHO-K1 cells after 6Gy irradiation were cultured at 37 °C for 1 h, and collected to process according to the steps in the method. As presented in Figure 3A, HCT116 cells and CHO-K1 cells only received 6Gy irradiation showed obvious DNA tail compared with the control cells. However, nuclear DNA in the cells treated with NOFD before irradiation was as complete as the control group. The data of the percentage of tail moment and tail DNA were presented in Figure 3B,C and the results indicated that NOFD exerted portent protective effect against IR-induced DNA damage.

### 2.4. NOFD Prevented IR-induced Apoptosis

Excessive ROS in cells has been demonstrated to attack the mitochondria and disrupt mitochondrial function, leading cells apoptosis and death [31,32]. In order to determine whether NOFD could inhibit cell apoptosis induced by IR, Annexin-V/PI double-staining method was conducted on HCT116 cells and CHO-K1 cells by a flow cytometry. Cells in the early stage of apoptosis were Annexin-V positive and PI negative, whereas cells in the late stage of apoptosis were both Annexin V and PI positive [33]. In Figure 4A, the results showed that the percentage of apoptosis cells acutely decreased after NOFD treatment in both HCT116 cells and CHO-K1 cells. The data graph was shown in Figure 4B. These data demonstrated that NOFD decreased the proportion of apoptotic cells evidently with regard to their respective controls.

### 2.5. NOFD Enhanced HIF-1α Protein Level and Target Genes Express

In order to assess whether NOFD acts on the HIF pathway, HIF-1 protein content and gene expression level were examined in this study. From the results, we could find that the amount of HIF-1 protein (Figure 5A) and gene expression (Figure 5B) in the cells increased significantly after administration with NOFD. Therefore, we could speculate that NOFD treatment prolonged the stabilization of HIF-1α protein in the cells under normoxia conditions.

Many of the downstream genes of HIF have important physiological functions, which may have a certain resistance to IR-induced damage. *VEGF* gene and *EPO* gene were proved dependence on HIF-1α and showed a certain degree of radiation protection [34,35]. In the study, these two genes were used as control genes to evaluate the regulation influence of NOFD on target genes. Furthermore, we chose dimethyloxalglycin (DMOG) as the control compound to compare the protective effect of NOFD against IR. DMOG, a PHD inhibitor, has been shown to increase the expression of HIF protein, activate downstream factors, and exert protective effect against hematopoietic damage and intestinal damage caused by IR. From the results, pretreatment NOFD increased the expression of VEGF gene as the same level as DMOG group in the cells (Figure 5C), which determined the effect of NOFD on the expression of HIF-1α target genes. Moreover, to determine the primary mechanism of NOFD against IR-induced damage, the mRNA level of HIF downstream genes, *Notch-1*, *Notch-2*, *Glut-1*, and *HO-1*, were examined by the same method. In Figure 5F, NOFD enhanced the level of *Notch-2* as the same level as the DMOG, while the level of *Notch1*, *Glut-1*, and *HO-1* genes in NOFD group had an increase compared to the control group, but had no apparently increasing like DMOG group (Figure 5E,G–H). These data suggested that NOFD could enhance the level of HIF-1α downstream genes, especially Notch2. This experiment provided new direction for researchers to explore the deep mechanism on the effect of NOFD against irradiation.

## 3. Discussion

Recently, although radiation therapy has made good progress in many diseases, the damage caused by ionizing radiation cannot be overlooked. IR-induced injury is most likely due to the large amount of ROS generated by ionizing radiation, which in turn can damage DNA and protein functions. Therefore, the removal of ROS remains an important direction in the research of radiation protection agents. In the body, HIF, as a transcription factor, plays an important role in metabolic activities. At the present stage, the study of HIF protein mostly stays in the study of chronic anemia disease and tumor therapy; while, studies have shown that the stability of HIF protein has a protective effect on the hematopoietic system injury and intestinal injury caused by ionizing radiation, but the research of HIF on the radiation protection is still very few. In this study, we synthesized NOFD, as an FIH inhibitor, and used cell models to study the effect NOFD on clearing the ROS produced by IR.

In our study, we found that NOFD could reduce the ROS levels enhanced by IR and relieve DNA damage and alleviate apoptosis in the cells. These results showed that NOFD exerted potent protective effect against IR-induced injury. Furthermore, the expression of HIF downstream genes was measured to explore the potential mechanism. HIF, as a transcription factor, regulates many downstream genes; many of these genes have important physiological functions, which may have a certain resistance on IR-induced damage. For example, VEGF is an important target gene of HIF, which has been already thought to be a peptide growth factor specific for vascular endothelial cells [36]. EPO is a glycoprotein hormone in the metabolic regulation, and the ability of EPO is to stimulate bone marrow erythrocyte production [37,38], and plays an initial role in the protection of hematopoietic damage. The role of VEGF and EPO genes in radiation protection is researched in numerous articles. In this study, we tested the relative expression of Notch-1, Notch-2, Glut-1, and HO-1 genes, and the results showed that the expression of Notch-2 gene was enhanced significantly.

Therefore, NOFD exerted potent protective effect on mitigating ROS levels and alleviating radiation-induced DNA damage and apoptosis in cells. In terms of mechanism, the notch2 signal pathway can be studied in more depth as new research content.

## 4. Materials and Method

### 4.1. N-oxalyl-d-phenylalanine (NOFD) Synthesis

Target compound NOFD were synthesized as reported in Figure 6 [39]. Dissolving D-phenylalanine methyl ester hydrochloride (2.2 g, 10.00 mmoL) in anhydrous CH_2_Cl_2_ (50 mL), Et_3_N (2.2 g, 11.00 mmoL) and methyloxalyl chloride (1.3 g, 22.00 mmoL) were added at 0 °C under nitrogen. The mixture was stirred for 5 min at 0 °C and then for 1 hr at room temperature. After the reaction is completed, the mixture was washed with sat.NaHCO_3_ (20 mL), and dried over Na_2_SO_4_, filtered and evaporated. The organic layer was dried to give the pale yellow crude product. Then the product was purified by column chromatography (petroleum ether 40–60: EtOAc 8:2 to 5:5) to yield 1.81 g (68%) of *N*-(Methyloxalyl)-D-phenylalanine methyl ester as a colorless oil. Then to a stirred solution of *N*-(Methyloxalyl)-D-phenylalanine methyl ester (1.81 g, 6.80 mmoL) in THF (15 mL) was added 1N NaOH (20 mL) and the resulting mixture was stirred for 2 h at room temperature. Then THF was evaporated and the resultant residue acidified with 1N HCl to pH = 2. The solution was extracted with EtOAc (2 × 20 mL). The combined organic extracts were dried over Na_2_SO_4_, filtered and evaporated in vacuo to afford 1.3 g (55%) of NOFD as a white solid.

### 4.2. Cell Culture and MTT Assay

HCT116 cells and CHO-K1 cells were cultured in basic DMEM medium supplemented with 10% fetal bovine serum and 1% penicillin and streptomycin antibiotic mixture at 37 °C in a humidified 5% CO_2_ atmosphere. The colorimetric 3-(4,5-dimethyl-2-thiazolyl)-2,5-diphenyl-2-H-tetrazolium bromide (MTT) is selected to determine the cell growth inhibition. After adding dimethyl sulfoxide (DMSO) to the cells 4 h MTT culture later, the color of the culture medium changes from yellow to purple. This change only occurs in living cells. The more the number of living cells, the heavier the color.

The cells were seeded with 96-well plates and cultured under normal conditions for 24 h, and then the old medium was discarded; NOFD was diluted to different concentrations with cell cultured medium, and then the fresh medium containing the different concentrations test compound (NOFD) would be replaced, continuing to incubate for 12 h. Post-administration Cells were incubated with MTT solution (0.5 mg/mL) for another 2 h, and the supernatant were removed carefully. Then, 150 μL DMSO was added to each well to dissolve formazan precipitate. The absorbance (A) of each well (six parallel wells for each sample) was measured at 490 nm using a Synergy HT Multi-Mode Microplate Reader (BioTek, Winooski, VT, USA). The inhibition rate was calculated using the following formula: Cell inhibition rate (%) = [(A(negative control)-A(test))/A(negative control)] × 100. Three parallel experiments were conducted (Figure 7). We choose the half IC_10_ concentration for the next experiments, considering its toxicity and negative effects on cells were negligible.

### 4.3. Apoptosis Assays

Apoptosis analysis was experimented in the HCT116 cells and CHO-K1 cells. Cells were incubated as described above in 6-well white plates. Before 6Gy irradiation, experiment cells were treated with 0.01 mg/mL NOFD and control cells were added with the same dosage of medium for 12 h. Then, cells were collected 24 h after irradiation to stain annexin V and PI according to the instructions for a BD apoptosis kit. The Data acquisitions were performed BD Accuri C6 instrument and analyzed using the BD Accuri C6 software (BD Bioscience, San Jose, CA, USA).

### 4.4. Comet Assay

The HCT116 cells and CHO-K1 cells were cultured in 12-well white plates as described above, and treated with 0.01 mg/Kg NOFD for 12 h before 6Gy irradiation. After IR, cells were incubated at 37 °C for 1 h, and then collected according to the density of 1 × 10^6^ cells per 1 mL PBS. High-melting-point agarose (0.8%) was first applied to the microscope slides. Low-melting-point agarose (0.6%) melted and was thermostated in a constant temperature of 37 °C. Then 30% of the cells and 70% of the low-melting-point agarose were mixed quickly and uniformly placed on the surface of the high-melting-point agarose. Drying slides were washed with double steamed water, and submerged in freshly prepared cold lysis buffer in 4 °C for 2.5 h. Next, we placed the slides in a horizontal gel electrophoresis unit filled with cold electrophoretic lysate for 30 min. After that, slides were electrophoresis at 30 V for 20 min electrophoresis fluid, and then neutralized in neutralization solution for 20 min. The slides were stained with ethidium bromide (2 µg/mL) for 10 s before observing. Last, slides were examined using a Nikon fluorescence microscope and DNA damage was estimated using the Comet Assay Software Project (CASP).

### 4.5. Immunofluorescence

Histone H2AX (γ-H2AX) was an important indicator responding to DNA double-strand breaks (DSBs) [40]. HCT116 and CHO-K1 cells were incubated in 24-well white plates and treated with 0.01 mg/Kg NOFD for 12 h before 6Gy irradiation. Incubated in 37 °C for 1 h after radiation, cells were washed with PBS for one time and then fixed with 4% paraformaldehyde for 20 min at 4 °C. After PBS wash for three times, cells were treated with 0.2% Triton X-100 for 15 min at room temperature, then washed with PBS and incubated with rabbit polyclonal γ-H2AX (phospho S139) antibody (dilution 1:1000, cat. No. ab2893; Abcam, Cambridge, MA, USA) overnight at 4 °C. After this, cells were incubated with goat anti-rabbit secondary antibody (dilution 1:2000, cat. No. ab6939; Abcam) for 2 h, and then washed with PBS for three times. Before observing, Nuclei were counterstained with DAPI (4,6-diamino-2-phenyl indole) (cat. No. C0065; Solarbio Science and Technology Co. Ltd., Beijing, China) for 5 min. We examined γ-H2AX foci of cells by monitoring fluorescence of the Cy3, and the images of cells were obtained using an AMG EVOS fluorescence microscope. In order to ensure randomness, nuclei selected for DAPI staining were random and then monitored for focus formation. For each group, at least 20 pictures were captured and used for analysis of γ-H2AX foci. In order to determine the ROS level in the cells, CHO-K1 and HCT116 cells were incubated in 12-well white plates for 24 h and cultured with 0.01 mg/mL NOFD for 12 h before IR. 5 µM 2,7-dichlordihydrofluorescein diacetate (DCFH-DA) was added in the medium for 20 min at 37 °C after radiation immediately. Removing the supernatant, cells were washed with PBS for three times, and then the fluorescence signal was measured at 488 nm with an Infinite F200 multimode plate reader.

### 4.6. Reverse Transcriptase-Polymerase Chain Reaction (rt-PCR)

The total RNA of HCT116 was extracted with TRIzol reagent (Life Technologies, Grand Island, NY, USA) 12 h after NOFD treatment. Reverse transcription was performed with a Revert Aid First Strand cDNA Synthesis Kit (Thermo Scientific, Waltham, MA, USA), according to the manufacturer’s instructions. All PCRs were conducted under an ABI 7500 Sequence Detection System and GAPDH (Thermo, Waltham, MA, USA) was used as control. Primers were designed for the *HIF-1α* gene: forward, 5′-gccctaacgtgttatctgtcg-3′ and reverse, 5′-ttgctccattccattctgttc-3′; the *VEGF* gene: forward, 5′-tgccaagtggtcccag-3′ and reverse, 5′-gtgaggtttgatccgc-3′; the *EPO* gene: forward,5′-tcccagacaccaaagttaatttcta-3′ and reverse, 5′-ccctgccagacttctacgg-3′; the *Notch-1* gene: forward, 5′-caatgtggatgcccgcagttgtg-3′ and reverse, 5′-cagcaccttggcggtctcgta-3′; the *Notch-2* gene: forward, 5′-tattgatgactgccctaaccaca-3′ and reverse, 5′-atagcctccattgcggttgg-3′; the *glut-1* gene: forward, 5′-ggttgtgccatactcatgacc-3′ and reverse, 5′-cagataggacatccagggtagc-3′; the *HO-1* gene:5′-gagtgtaaggacccatcgga-3′ and reverse, 5′-gccagcaacaaagtgcaag-3′.

### 4.7. Western-Blot Analysis

HCT116 cells were collected 12 h after treated 0.01 mg/mL, 0.05 mg/mL, and 0.1 mg/mL NOFD respectively, and kept at 4 °C for 30 min with RIPA. The control cells after exchanged with new medium were cultured for the same time. Total protein lysates of all the cells were obtained by centrifuged at 4 °C for 15 min for immunoblot analysis. The cell lysates were separated by 10% SDS-PAGE and transferred onto PVDF membranes. Proteins were then prepared to immunoblot.

### 4.8. Statistical Analysis

The results were performed as mean ± SD. The histogram was made using the Graphpad software, and the line chart was made using the Origin software. Statistical analysis was performed by one-way ANOVA using SPSS 19.0 (SPSS Inc, Chicago, IL, USA). Statistical significance was taken as *p* < 0.05, *p* < 0.01 or *p* < 0.001.

## Figures and Tables

**Figure 1 ijms-20-00037-f001:**
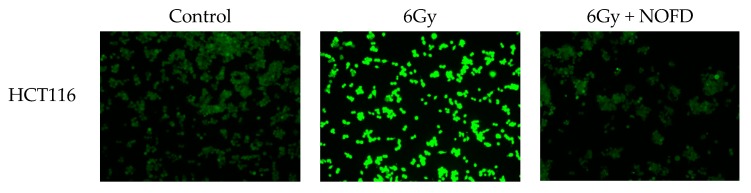
NOFD reduces ROS level. Fluorescence images of HCT116 and CHO-K1 cells from the control, 6Gy, and 6Gy + NOFD groups.

**Figure 2 ijms-20-00037-f002:**
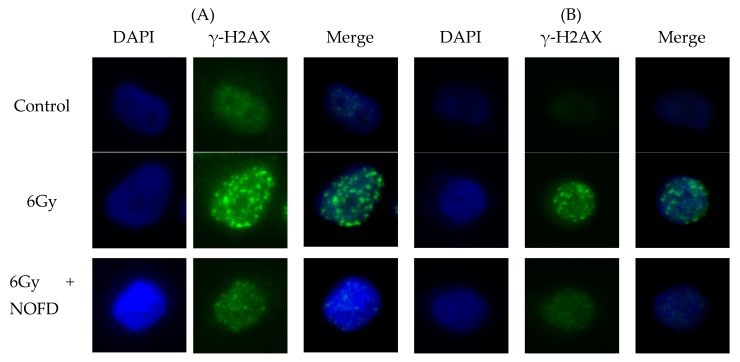
NOFD reduces fluorescence signal of γ-H2AX in HCT116 and CHO-K1 cells (**A**) The γ-H2AX representative expression (×40) in CHO-K1 cells 1 h after 6Gy irradiation. (**B**) The γ-H2AX representative expression (×40) in HCT116 cells 1 h after 6Gy irradiation.

**Figure 3 ijms-20-00037-f003:**
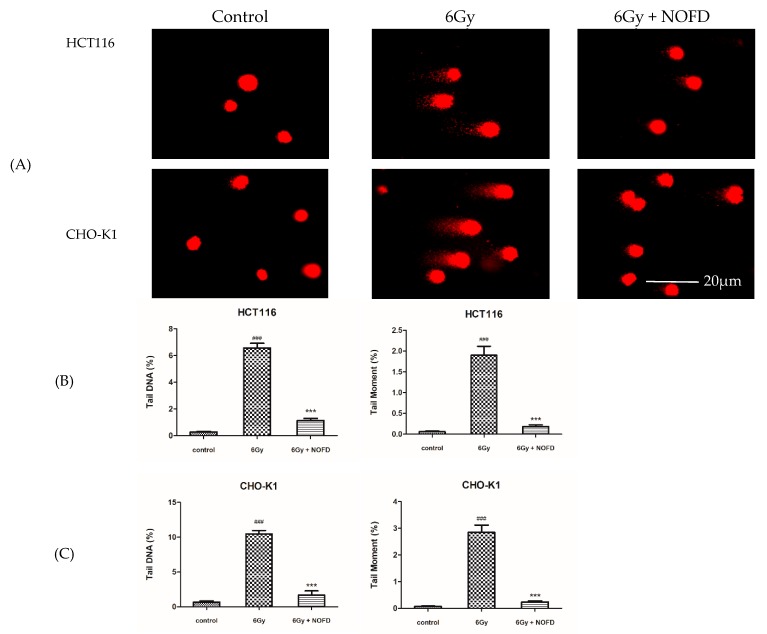
NOFD ameliorates IR-induced DSBs in HCT116 and CHO-K1 cells. (**A**) Representative comet images from different groups. The column graph expresses the percentage of tail DNA and tail moment in HCT116 cells (**B**) and CHO-K1 cells (**C**) (mean ± SD, *n* = 6); ### *p* < 0.001 vs. control; *** *p* < 0.001 vs. 6Gy.

**Figure 4 ijms-20-00037-f004:**
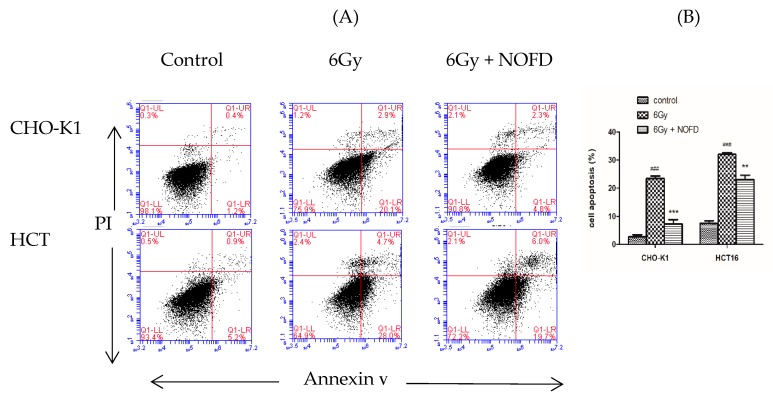
NOFD reduces IR-induced apoptosis. (**A**) The representative FACS plots shows the percentages of cell apoptosis. (**B**) The bar shows the percentage of apoptosis (mean ± SD, *n* = 3). ### *p* < 0.001 vs. control; ** *p* < 0.01 and *** *p* < 0.001 vs. 6Gy group.

**Figure 5 ijms-20-00037-f005:**
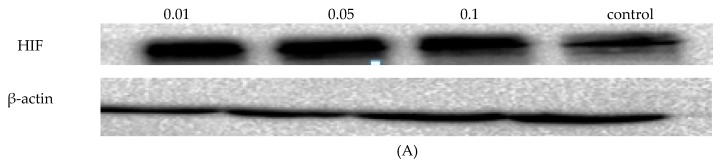
NOFD up-regulates HIF-1α protein level and the expression of related genes in HCT16 cells. (**A**) The bar shows the level of HIF-1α protein in HCT116 cells. (**B**–**H**) The bar graphs show the mRNA expression of HIF-1α *VEGF*, *EPO*, *Notch-1*, *Notch-2*, *Glut-1*, and *HO-1* in HCT116 cells (mean ± SD, *n* = 3). * *p* < 0.05, ** *p* < 0.01 and *** *p* < 0.001 vs. control.

**Figure 6 ijms-20-00037-f006:**
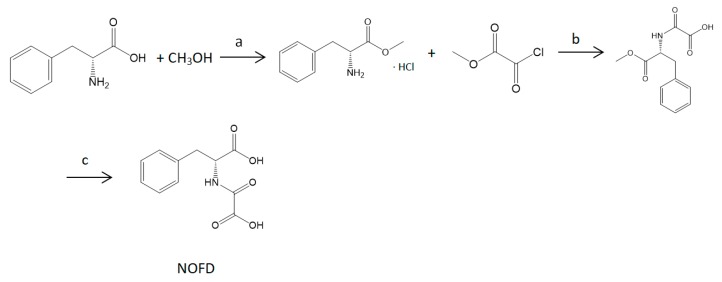
Reagents and conditions: (a) SOCl2, 0 °C, 4 h; (b) NH3 (aq), DCM, 1 h; (c) NaOH, 2 h.

**Figure 7 ijms-20-00037-f007:**
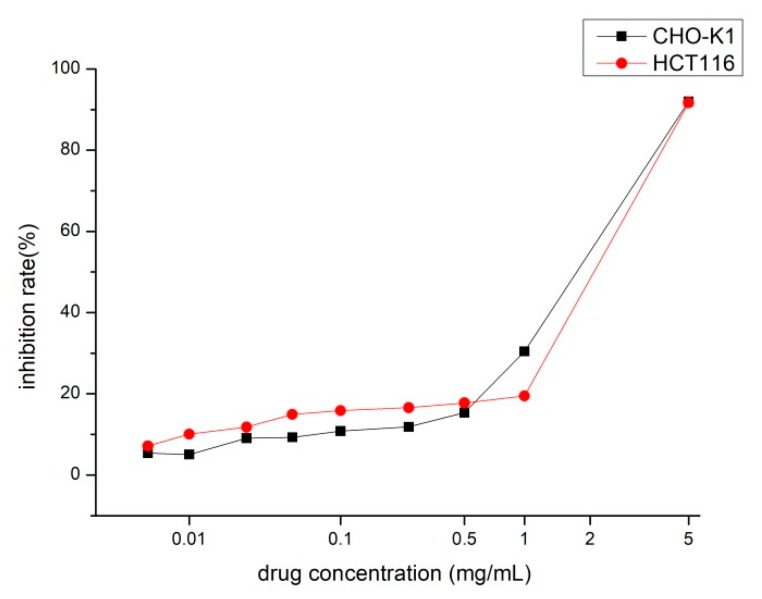
The effect of NOFD on inhibition rate of HCT116 cells and CHO-K1 cells.

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
