# Peer review of "Radioprotective Activity and Preliminary Mechanisms of N-oxalyl-d-phenylalanine (NOFD) In Vitro"

_ijms, 2018, doi:10.3390/ijms20010037_

Reviewer 1 Report

Interesting scientific data about the role of HIF stability in cell radioresistance is presented in the article. This study showed that pharmacological HIF blockage have radiation protective effect. Section of statistic methods is absent that must be include in the article.

The article is recommended to publication.

Author Response

point 1 :Remoulding the statement in the material description section

response 1 :The retouching section has been marked in the new attachment file.

Reviewer 2 Report

Meng et al. analyzed the effect of an inhibitor, NOFD, of inhibitor proteins if HIF-1 and could show reduced radiosensitivity of cells after treatment with NOFD. Although the assays clearly show reduced radiosensitivity one essential experiment is missing.

Major:

The authors must show that the effect is really due to prolonged stabilization of HIF-1 by analysis of the nuclear levels of HIF-1.

They investigated expressions of genes which should be targets of HIF-1. However, they did not give any reference for dependence of the four genes on HIF-1. To prove their regulation by HIF-1 ate least two of those genes with proven dependence on HIF-1 should be used as control (Semenza, Biochem. Pharmacol., 64 (2002) 993-998 or Moeller et al. Cancer Cell 5,5 (2004) 429-441) or the Hypoxia Signaling Pathway RT2 Profiler PCR Array by Qiagen should be used.

Minor:

According to Materials & Methods the time intervals between NOFD treatment irradiation and harvesting of cells were different in the various assays. This should be written more clearly and explained.

Author Response

point 1:Overall description of the article

response 1:The description of the article has been revised in the attachment.

point 2:The authors must show that the effect is really due to prolonged stabilization of HIF-1 by analysis of the nuclear levels of HIF-1.

response 2: Western-blotting and rt-PCR experiments were added to assess the changes in the expression of HIF proteins and genes in cells after NOFD administration.

point 3:They investigated expressions of genes which should be targets of HIF-1. However, they did not give any reference for dependence of the four genes on HIF-1. To prove their regulation by HIF-1 ate least two of those genes with proven dependence on HIF-1 should be used as control (Semenza, Biochem. Pharmacol., 64 (2002) 993-998 or Moeller et al. Cancer Cell 5,5 (2004) 429-441) or the Hypoxia Signaling Pathway RT2 Profiler PCR Array by Qiagen should be used

response 3: According to the literature, adding VEGF and EPO genes as control genes.

point 4:According to Materials & Methods the time intervals between NOFD treatment irradiation and harvesting of cells were different in the various assays. This should be written more clearly and explained.

response 4:An explanation of the experimental time is shown in the attachment.

Round  2

Reviewer 2 Report

Meng et al. adressed all suggestions of the reviewer.

However, it is somewhat astonishing that results of all suggested experiments could be shown within less than a week.

Was every experiment done only once or, for example, does the depicted Western Blot show the representative result of three independent experiments?

It is only written for the MTT assay that "three parallel experiments were conducted".

Author Response

I have to admit that the experiments I have done are in parallel and that the article has been changed in the established format.

point 1:Have I completed the parallel experiment of WB?

response1:I completed the parallel experiment of WB. Because the research team is conducting experiments at the molecular level, the experimental results of WB already exist, causing me to respond to the experimental changes more quickly.

point 2:Questions about MTT experiments

response 2: The MTT experiment consisted of 6 replicate wells and three sets of parallel experiments, and the results have been shown in the article.